# Histopathology Image Report Generation by Vision Language Model with Multimodal In-Context Learning

**Liu, Shih-Wen**                                    NE6134079@GS.NCKU.EDU.TW
*National Cheng Kung University, Taiwan*
**Fan, Hsuan-Yu**                                    P76124702@GS.NCKU.EDU.TW
*National Cheng Kung University, Taiwan*
**Chu, Wei-Ta** [ID]                                    WTCHU@GS.NCKU.EDU.TW
*National Cheng Kung University, Taiwan*
**Yang, Fu-En**                                    FREDY@NVIDIA.COM
*NVIDIA Research*
**Wang, Yu-Chiang Frank**                                    FRANKWANG@NVIDIA.COM
*NVIDIA Research*

**Editors:** Accepted for publication at MIDL 2025

## Abstract

Automating medical report generation from histopathology images is a critical challenge requiring effective visual representations and domain-specific knowledge. Inspired by the common practices of human experts, we propose an in-context learning framework called PathGenIC that integrates context derived from the training set with a multimodal in-context learning (ICL) mechanism. Our method dynamically retrieves semantically similar whole slide image (WSI)-report pairs and incorporates adaptive feedback to enhance contextual relevance and generation quality. Evaluated on the HistGen benchmark, the framework achieves state-of-the-art results, with significant improvements across BLEU, METEOR, and ROUGE-L metrics, and demonstrates robustness across diverse report lengths and disease categories. By maximizing training data utility and bridging vision and language with ICL, our work offers a solution for AI-driven histopathology reporting, setting a strong foundation for future advancements in multimodal clinical applications.

**Keywords:** Multimodal In-Context Learning, Medical Report Generation, Histopathology Images, Vision-Language Models, HistGen Benchmark.

## 1. Introduction

Histopathology reports are important diagnostic tools, providing detailed interpretations of histopathological findings that directly influence patient care. Composing a histopathology report demands significant expertise and is very time-consuming. Therefore, automating this process holds immense potential to enhance diagnostic efficiency and reduce workloads. Yet, this task is far from trivial, as it requires understanding the intricate visual features of histopathology images and generating accurate and structured documents.

Recent advancements in medical AI have significantly improved the integration of visual and textual data for histopathology report generation. Models like HistGen (Guo et al., 2024) and WsiCaption (Chen et al., 2024) have made strides in bridging the gap between whole slide images (WSIs) and histopathology reports through multimodal approaches.

Similarly, models like Quilt-LLaVA (Seyfioglu et al., 2025) and LLaVA-Med (Li et al., 2023) integrate medical imaging and language models for broader applications, including Visual Question Answering (VQA) and histopathology report generation. Despite these advances, these medical models overlook utilizing the similarity between a test image and the images (and their associated histopathology reports) in the training datasets.

Taking similar images and their associated reports from previously confirmed cases as the reference is a common practice for expert pathologists to make a report efficiently and ensure report quality. Reviewing similar cases can facilitate maintaining diagnostic accuracy and consistency (Doe and Smith, 2023). They also rely on the Diagnosis Learning Cycle (Branson et al., 2021), a conceptual framework designed to enhance diagnostic performance through feedback, reflection, and continuous learning, emphasizing the importance of learning from past experiences. Inspired by this, we propose to retrieve similar images and their associated reports from the database and then take them as important clues for generating a report of the test image. We employ a fine-tuned histopathology-specific vision language model (VLM) and investigate prompting techniques to generate accurate histopathology reports.

We evaluate our framework on the HistGen benchmark, achieving state-of-the-art performance across BLEU, METEOR, and ROUGE-L metrics. We also test the proposed components separately and study performance obtained based on different settings. In summary, our contributions include:

- Introducing an expert-inspired framework with multimodal in-context learning to utilize information from the training dataset to generate better histopathology reports.

- Fine-tuning a histopathology-specific VLM to enhance its capabilities in generating histopathology reports.

- Achieving state-of-the-art results on the HistGen benchmark and demonstrating the robustness and efficacy across various configurations.

## 2. Related Works

**Medical Report Generation.** Medical report generation has received significant attention recently. Models like HistGen (Guo et al., 2024) and WsiCaption (Chen et al., 2024) have demonstrated success in generating histopathology reports by bridging WSIs and textual data through advanced encoding and generative approaches. HistGen employs local-global feature encoding to enhance contextual coherence in reports. WsiCaption uses a multiple-instance generative model to produce detailed clinical descriptions. Vision-language models like Quilt-LLaVA (Seyfioglu et al., 2025) and LLaVA-Med (Li et al., 2023) extend these capabilities to broader biomedical applications. LLaVA-Med adapts the LLaVA framework to handle diverse medical tasks, including report generation and VQA, by leveraging curriculum learning to align visual and textual data. Quilt-LLaVA focuses on histopathology-specific applications, aligning representations of WSIs and text using histopathology-specific datasets like Quilt-Instruct. While these models achieve automatic report generation, they often overlook contextual information from training datasets, diverse histopathology categories, or the common practices of expert pathologists.

**In-Context Learning and Retrieval-Augmented Generation.** In-context learning (ICL) has emerged as a promising technique for improving model adaptability without retraining. Pioneered by models like GPT-3 (Brown et al., 2020), ICL enables models to work on tasks by incorporating task-specific examples directly into the input prompts. Retrieval-augmented generation (RAG) (Lewis et al., 2020) is one of the ICL strategies that embodies this concept by utilizing retrieved data to increase model adaptability. While ICL and RAG have shown success in general-purpose language models, their application to medical AI remains limited. Existing multimodal models like Quilt-LLaVA and LLaVA-Med focus on directly encoding visual and textual data but lack robust mechanisms for retrieving and leveraging similar examples for better generation. Our work attempts to bridge the gaps by incorporating ICL into a report generation framework based on context derived from similar training examples, category-specific guidelines, and structured feedback.

## 3. Methodology

### 3.1. Base Model

Figure 1 illustrates the proposed report generation framework. We take Quilt-LLaVA as the base model, which has been trained on diverse histopathology datasets for VQA tasks. It was trained based on histopathology image patches and was dedicated to VQA. To more effectively extract features from the entire histopathology image and focus on report generation, we replace its visual encoder with Vision Transformer Large (ViT-L) provided in HistGen (Guo et al., 2024). The ViT-L has been extensively pre-trained using the DINOv2 technique and is used to extract WSI[1] features $\mathbf{F}_{patch} \in \mathbb{R}^{n \times d}$, where $n$ is the number of patches in a WSI and $d$ is the feature dimension.

The extracted patch-based features $\mathbf{F}_{patch}$ and $m$ learnable query tokens $\mathbf{Q} \in \mathbb{R}^{m \times d}$ are processed by transformer blocks. The goal is to consider the context between patches to form holistic WSI features:

$$\hat{\mathbf{H}} = \text{TransformerBlock}(\mathbf{Q}, \mathbf{F}_{patch}), \tag{1}$$

where $\hat{\mathbf{H}} \in \mathbb{R}^{m \times d}$ represents the $m$ contextualized tokens derived from $\mathbf{Q}$. Referring to Figure 1, the contextualized tokens $\hat{\mathbf{H}}$ are processed by a projector to be the embeddings more appropriate to a VLM, and the results are denoted as $\mathbf{H} \in \mathbb{R}^{m \times d}$ in the following. Notice that the number of patches $n$ in different WSIs may differ. But we always represent a WSI by the $m$ contextualized tokens $\mathbf{H}$.

The tokens $\mathbf{H}$ are then passed to a VLM along with a text prompt $\mathbf{P}_{text}$ to generate a histopathology report $\mathbf{Y}_{gen}$:

$$\mathbf{Y}_{gen} = \text{VLM}(\mathbf{H}, \mathbf{P}_{text}). \tag{2}$$

The text prompt $\mathbf{P}_{text}$ (shorten version) for the baseline model is: `What are the diagnostic findings in the image? Provide a professional, accurate, and well-structured histopathology report for it.`

---

1. All histopathology data are represented in whole slide images in this work. Therefore, we use histopathology image and whole slide image interchangeably.

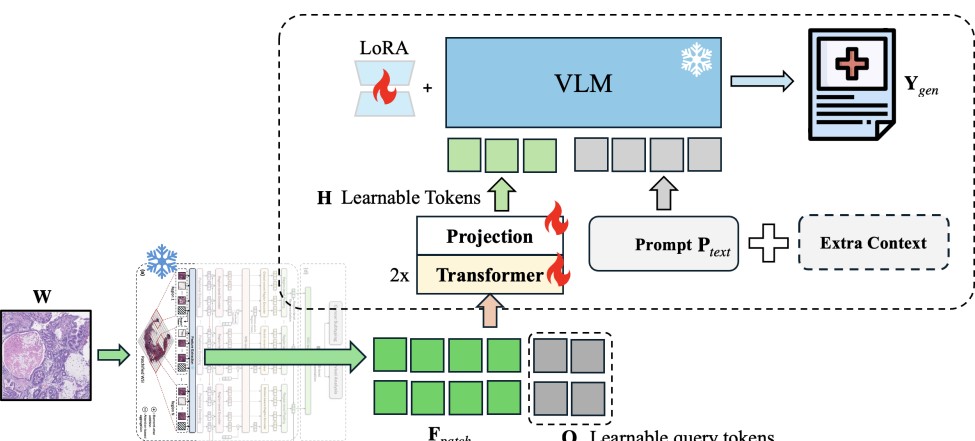

Figure 1: Overview of the proposed framework. WSI patch features $\mathbf{F}_{patch}$ and the learnable tokens $\mathbf{Q}$ are jointly processed by a transformer to get holistic WSI features $\mathbf{H}$. The processed $\mathbf{H}$ with the text prompts $\mathbf{P}_{text}$ are then fed to a VLM to generate a report $\mathbf{Y}_{gen}$. Our main contribution is enriching prompts with extra context. The components with flame symbols mean that we need to train or fine-tune the parameters, and the ones with snowflake symbols mean that we adopt the parameters pre-trained by existing works and these parameters are frozen in the training process.

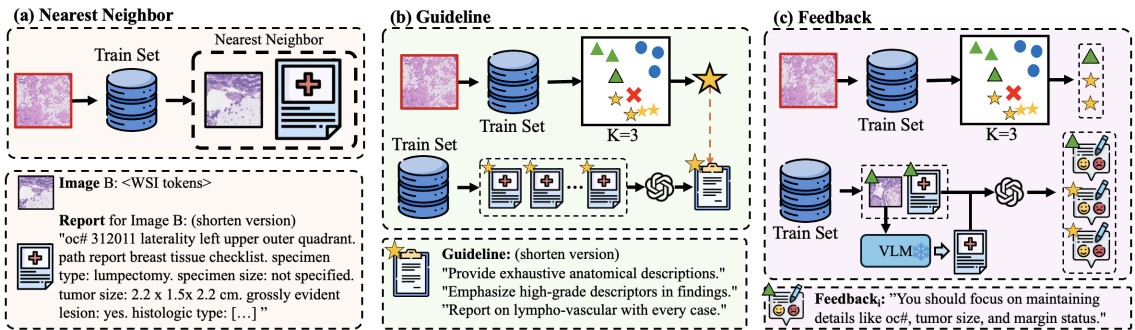

Figure 2: Illustrations of three different clues for in-context learning.

## 3.2. In-Context Learning

Based on the base model described above and the framework shown in Figure 1, we propose to extract extra context from the training dataset to facilitate better report generation. The three in-context learning mechanisms are nearest neighbor, category guideline, and feedback. Figure 2 illustrates these three different mechanisms.

**Nearest Neighbor.** In this mechanism, we retrieve the WSI-report pair that is closest to the test WSI from the training set. We calculate the cosine similarity between the WSI tokens of the test image $\mathbf{H}_{test}$ and every WSI in the training set $\mathbf{H}_{train}$. The image tokens of

the WSI most similar to $\mathbf{H}_{test}$, denoted as $\mathbf{H}_{ret}$, are appended to the test WSI tokens, i.e., $\mathbf{H}_{nn} = (\mathbf{H}_{test}, \mathbf{H}_{ret})$. The histopathology report $\mathbf{Y}_{ret}$ corresponding to $\mathbf{H}_{ret}$ is appended to the text prompt, i.e., $\mathbf{P}_{nn} = (\mathbf{P}_{text}, \mathbf{Y}_{ret})$.

Given $\mathbf{H}_{nn}$ and $\mathbf{P}_{nn}$, we guide the VLM to generate better histopathology reports with the visually similar WSI tokens and their corresponding report as a reference, i.e.,

$$\mathbf{Y}_{gen} = \text{VLM}(\mathbf{H}_{nn}, \mathbf{P}_{nn}). \tag{3}$$

**Category Guideline.** The WSIs in the HistGen benchmark are classified into 32 categories according to disease types. Each WSI is labeled with a disease type. Considering cues from the disease category that is closest to the test WSI may benefit report generation. To do this, we compare the test WSI tokens $\mathbf{H}_{test}$ with WSIs in the training set. We check the disease types of the $K$ WSIs that are most similar to $\mathbf{H}_{test}$. The major category of these $K$ WSIs is then determined. Let's denote it as the category $C$.

There are multiple WSIs and their associated reports in the category $C$. We then ask GPT-4o[2] to review these medical reports and generate a representative guideline $\mathbf{T}_{rep}$. With this clue, we finally ask a VLM to generate a report of the test WSI by giving it $\mathbf{H}_{test}$, $\mathbf{T}_{rep}$, and $\mathbf{P}_{text}$, i.e.,

$$\mathbf{Y}_{gen} = \text{VLM}(\mathbf{H}_{test}, \mathbf{T}_{rep}, \mathbf{P}_{text}). \tag{4}$$

**Feedback.** Generated reports are usually not perfect. Considering the difference between a generated report and its perfect counterpart may provide clues to enhance report generation. Specifically, we first generate a report for each WSI in the training set by the base model. For each WSI, we then ask GPT-4o to compare the generated report with the corresponding truth report and generate feedback about how to improve the generated version. For each WSI $W_i$, we thus have the corresponding feedback $\mathbf{B}_i$. We believe that report generation can benefit from feedback, as emphasized in the Diagnosis Learning Cycle.

During testing, we find the $K$ WSIs most similar to the test WSI and obtain the $K$ feedback. A VLM is then asked to generate a report for the test WSI by giving the $K$ feedback and $K$ retrieved WSI tokens, i.e.,

$$\mathbf{Y}_{gen} = \text{VLM}(\mathbf{H}_{test}, \{\mathbf{B}_j\}_{j=1}^K, \mathbf{P}_{text}). \tag{5}$$

The aforementioned three contexts, including nearest neighbor, category guideline, and feedback, can be used for in-context learning separately. We can also jointly use them by inputting all elements mentioned in eqn. (3), eqn. (4), and eqn. (5). In the following, the full version of the proposed model is called **PathGenIC**, standing for HistoPathology report Generation with In-Context learning.

### 3.3. Loss Function and Training

The model is trained based on the cross-entropy loss to align the generated report $\mathbf{Y}_{gen}$ with the ground truth report $\mathbf{Y}_{true}$. During training, the components with the flame symbols in Figure 1 are trained to complete this framework. The fine-tuned parts include the transformer blocks to process WSI patch tokens and learnable query tokens and the LoRA (Hu et al., 2022) component that adapts the VLM, i.e., Quilt-LLaVA, to generate reports.

---

2. The exact version is gpt-4o-2024-08-06.

Table 1: Performance comparison of different methods on the HistGen benchmark.

| Feature Extractor | Methods | BLEU-1 | BLEU-2 | BLEU-3 | BLEU-4 | METEOR | ROUGE-L | fact$_{ENT}$ |
|---|---|---|---|---|---|---|---|---|
| ResNet50 | **Show&Tell** (Vinyals et al., 2015) | 0.249 | 0.099 | 0.047 | 0.025 | 0.086 | 0.165 | |
| | **UpDownAttn** (Anderson et al., 2017) | 0.250 | 0.115 | 0.065 | 0.043 | 0.096 | 0.180 | |
| | **Transformer** (Vaswani et al., 2017) | 0.249 | 0.114 | 0.065 | 0.042 | 0.095 | 0.176 | |
| | **M2Transformer** (Cornia et al., 2020) | 0.250 | 0.115 | 0.065 | 0.042 | 0.095 | 0.180 | |
| | **R2Gen** (Chen et al., 2020) | 0.240 | 0.105 | 0.058 | 0.036 | 0.089 | 0.177 | |
| | **R2GenCMN** (Chen et al., 2021) | 0.225 | 0.095 | 0.047 | 0.022 | 0.094 | 0.151 | |
| CTransPath | **Show&Tell** (Vinyals et al., 2015) | 0.262 | 0.126 | 0.071 | 0.043 | 0.094 | 0.184 | |
| | **UpDownAttn** (Anderson et al., 2017) | 0.240 | 0.139 | 0.090 | 0.063 | 0.100 | 0.201 | |
| | **Transformer** (Vaswani et al., 2017) | 0.271 | 0.165 | 0.112 | 0.082 | 0.113 | 0.227 | |
| | **M2Transformer** (Cornia et al., 2020) | 0.259 | 0.160 | 0.108 | 0.076 | 0.103 | 0.218 | |
| | **R2Gen** (Chen et al., 2020) | 0.237 | 0.135 | 0.085 | 0.054 | 0.086 | 0.205 | |
| | **R2GenCMN** (Chen et al., 2021) | 0.211 | 0.098 | 0.054 | 0.033 | 0.079 | 0.158 | |
| DINOv2 ViT-L | **Show&Tell** (Vinyals et al., 2015) | 0.189 | 0.094 | 0.056 | 0.039 | 0.070 | 0.165 | |
| | **UpDownAttn** (Anderson et al., 2017) | 0.320 | 0.206 | 0.147 | 0.112 | 0.131 | 0.271 | |
| | **Transformer** (Vaswani et al., 2017) | 0.382 | 0.266 | 0.200 | 0.157 | 0.162 | 0.316 | |
| | **M2Transformer** (Cornia et al., 2020) | 0.321 | 0.213 | 0.152 | 0.112 | 0.131 | 0.266 | |
| | **R2Gen** (Chen et al., 2020) | 0.274 | 0.166 | 0.107 | 0.071 | 0.102 | 0.234 | |
| | **HistGen** (Guo et al., 2024) | 0.413 | 0.297 | 0.229 | 0.184 | 0.182 | 0.344 | |
| DINOv2 ViT-L | **Base (Ours)** | **0.411** | **0.290** | **0.222** | **0.178** | **0.184** | **0.336** | 0.445 |
| | **PathGenIC (Ours)** | **0.431** | **0.313** | **0.243** | **0.196** | **0.197** | **0.357** | 0.462 |

## 4. Experiments

### 4.1. Implementation Details

**Benchmark: HistGen Dataset.** We use the HistGen dataset (Guo et al., 2024) for evaluating report generation. It contains 7,690 WSIs and their corresponding histopathology reports collected from TCGA, spanning 32 disease categories. It is divided into training (80%), validation (10%), and test (10%) subsets, resulting in 6,152 training samples, 769 validation samples, and 769 test samples.

To fairly compare with existing methods on the HistGen dataset, we adopt BLEU, METEOR, and ROUGE-L as the evaluation metrics. These metrics collectively measure lexical similarity, semantic relevance, and structural coherence between the generated and ground-truth reports. However, they were proposed from the natural language processing perspective and may not well reflect domain entities or inferential consistency (Miura et al., 2021). To enhance the evaluation, we further show performance in terms of Exact Entity Match Reward (fact$_{ENT}$) proposed in (Miura et al., 2021), which captures the completeness of a generated report by measuring its coverage of entities.

**Experiments Setup.** We use a batch size of 8 and train the model for 20 epochs. The Adam optimizer is employed with an initial learning rate of $1 \times 10^{-4}$, which follows a cosine reduction schedule to zero. These settings are designed empirically to ensure performance convergence. Quilt-LLaVA is selected as the base model because it is pre-trained on a big histopathology image dataset for the VQA task. This provides a good foundation for us to extend to histopathology report generation. All experiments are conducted on 2 NVIDIA RTX3090 GPU with 24 GB memory. The implementation is based on PyTorch. To calculate the value of fact$_{ENT}$, we adopt BioBERT-v1.1 as the named entity recognition model.

### 4.2. Performance of Report Generation

Table 1 shows a performance comparison between different methods on the HistGen benchmark. As can be seen, both our base model and the PathGenIC model outperform existing

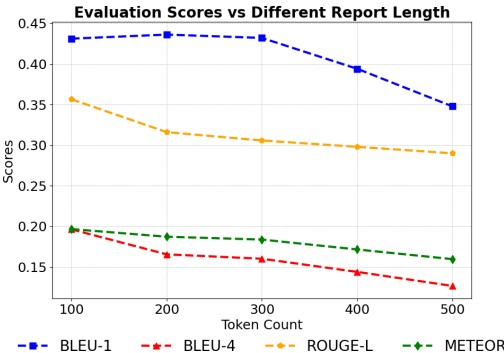

Figure 3: Performance across varying sequence lengths (100 to 500 tokens).

Table 2: Performance variations when different numbers of nearest neighbors ($K$) in WSI-report pair retrieval.

| Method | BLEU-1 | BLEU-4 | METEOR | ROUGE-L |
|--------|--------|--------|--------|---------|
| $K$=1 | 0.431 | 0.195 | 0.196 | 0.356 |
| $K$=3 | 0.431 | 0.196 | 0.197 | 0.357 |
| $K$=5 | 0.428 | 0.192 | 0.195 | 0.352 |

methods. With the in-context learning mechanism, the PathGenIC method improves the base model, showing the benefits brought by contextual clues. We also show the values of $fact_{ENT}$ obtained by our methods in the rightmost column, while the values of other methods are not available.

**Analysis of Report Length.** The results of all methods shown in Table 1 are obtained based on only the first 100 tokens of the generated reports, roughly 80 to 90 words. To understand the proposed PathGenIC more deeply, we study how the report's length influences performance. Figure 3 shows that most metrics slightly decline as the report's length increases, but the overall quality remains high. Interestingly, we observe a slight improvement in the BLEU-1 scores as the report length increases from 100 to 200 tokens. This suggests that slightly longer reports may enhance the likelihood of word matches between the generated and ground truth reports, possibly due to a more extensive context allowing for better word choice predictions. As the report length increases more, the occurrence of less common words rises and makes a downward trend in metrics.

**Ablation Study of #Retrieved WSIs.** We retrieve the top $K$ WSIs and their associated reports as the reference in the category guideline and the feedback mechanisms. We evaluate how different $K$'s influence performance in Table 2. The results show that using $K = 3$ gives the best performance. Using $K = 1$ offers comparable performance with slightly reduced effectiveness. This reduction can be attributed to the instability of clues when fewer cases are considered. Conversely, using $K = 5$ introduces noise into the system and negatively affects the model's performance.

Table 3: Performance variations when different components are applied.

| Method | BLEU-1 | BLEU-4 | METEOR | ROUGE-L |
|---|---|---|---|---|
| Base Model | 0.411 | 0.178 | 0.184 | 0.336 |
| + NN | 0.428 | 0.193 | 0.194 | 0.354 |
| + NN + Guideline | 0.429 | 0.195 | 0.195 | 0.355 |
| + NN + Feedback | 0.429 | 0.193 | 0.195 | 0.354 |
| **+ NN + Guideline + Feedback** | **0.431** | **0.196** | **0.197** | **0.357** |

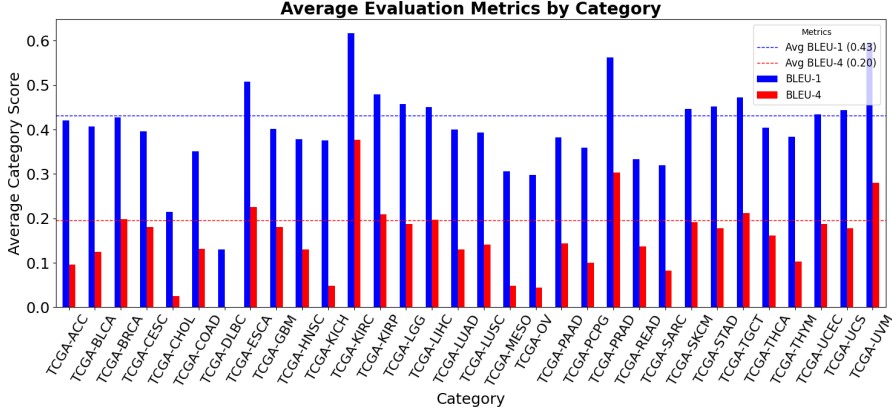

Figure 4: BLEU scores across the 32 disease categories.

**Ablation Study of Components.** We evaluate how different components influence performance in Table 3. By comparing the first two rows, we see that the evident performance improvement can be obtained when in-context learning is applied, e.g., the BLEU-1 score improved from 0.411 to 0.428, and the ROUGE-L score improved from 0.336 to 0.354. The best results are observed when all components are integrated.

**Performance on Different Diseases.** We analyze BLEU scores across 32 disease categories to deepen our understanding of model performance. Figure 4 presents a bar plot showing BLEU-1 and BLEU-4 scores for different categories, indicating that different diseases pose distinct challenges for our model.

### 4.3. Sample Results

Figure 5 and Figure 6 show sample results of generated reports and their corresponding ground truths, respectively. In both figures, major differences between the ground truth and the generated reports are underlined, and the performance in terms of BLEU, ME-TEOR, and ROUGE-L is shown. Two observations can be made from these two samples. First, although these metrics may not be the most appropriate metrics to evaluate medical report generation, we clearly can see that better generation (Figure 5) gives higher values. Second, the main reason for bad results (Figure 6) is not generating incorrect information or irrelevant information but rather missing a few key items in the report.

Both good and bad samples belong to the TCGA-KIRC subset. Based on KNN, both were correctly classified to get TCGA-KIRC guidelines, and the retrieved feedback highlighted several important aspects. However, we found that the main reason for poor per-

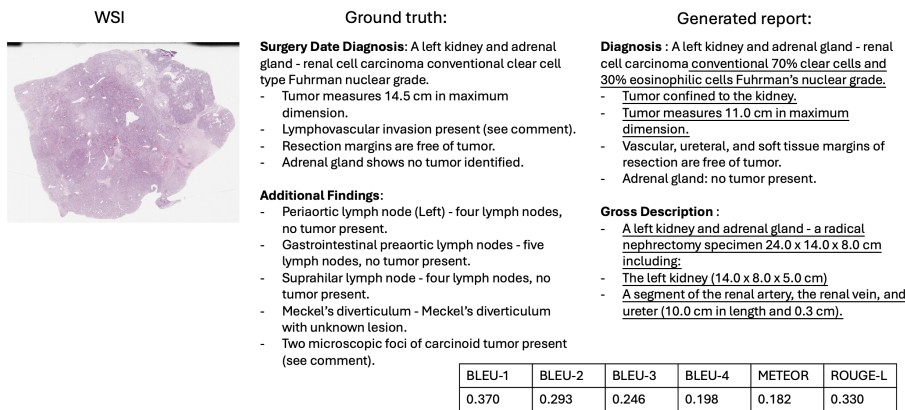

WSI  Ground truth:  Generated report:

**Clinical Diagnosis History**: Left renal mass.
**Specimens Submitted**: SP left kidney. SP portion of left 11th rib. SP peri-hilar tissue. SP left adrenal gland.
**Diagnosis**:
Kidney left radical nephrectomy - renal cell carcinoma clear cell type nuclear grade II/IV.
- The tumor's greatest diameter is 4.5 cm.
- The tumor is confined within the renal capsule.
- No invasion of the renal vein is identified.
- All surgical margins are free of tumor.
- The non-neoplastic kidney shows focal chronic inflammation only.

**Clinical Diagnosis History**: Left renal mass.
**Specimens Submitted**: 1 SP left kidney. 2 SP paracaval lymph nodes.
**Diagnosis**:
1. Kidney left nephrectomy - renal cell carcinoma clear cell type nuclear grade II/IV.
- The pattern of growth is acinar.
- The tumor's greatest diameter is 5.5 cm.
- The tumor is confined within the renal capsule.
- No invasion of the renal vein is identified.
- All surgical margins are free of tumor.
- The non-neoplastic kidney is unremarkable.
- The adrenal gland is not submitted.

| BLEU-1 | BLEU-2 | BLEU-3 | BLEU-4 | METEOR | ROUGE-L |
|--------|--------|--------|--------|--------|---------|
| 0.760 | 0.690 | 0.640 | 0.603 | 0.403 | 0.710 |

Figure 5: A sample result of the generated report and its corresponding ground truth. The generated result is relatively better.

WSI  Ground truth:  Generated report:

**Surgery Date Diagnosis**: A left kidney and adrenal gland - renal cell carcinoma conventional clear cell type Fuhrman nuclear grade.
- Tumor measures 14.5 cm in maximum dimension.
- Lymphovascular invasion present (see comment).
- Resection margins are free of tumor.
- Adrenal gland shows no tumor identified.

**Additional Findings**:
- Periaortic lymph node (Left) - four lymph nodes, no tumor present.
- Gastrointestinal preaortic lymph nodes - five lymph nodes, no tumor present.
- Suprahilar lymph node - four lymph nodes, no tumor present.
- Meckel's diverticulum - Meckel's diverticulum with unknown lesion.
- Two microscopic foci of carcinoid tumor present (see comment).

**Diagnosis** : A left kidney and adrenal gland - renal cell carcinoma conventional 70% clear cells and 30% eosinophilic cells Fuhrman's nuclear grade.
- Tumor confined to the kidney.
- Tumor measures 11.0 cm in maximum dimension.
- Vascular, ureteral, and soft tissue margins of resection are free of tumor.
- Adrenal gland: no tumor present.

**Gross Description** :
- A left kidney and adrenal gland - a radical nephrectomy specimen 24.0 x 14.0 x 8.0 cm including:
- The left kidney (14.0 x 8.0 x 5.0 cm)
- A segment of the renal artery, the renal vein, and ureter (10.0 cm in length and 0.3 cm).

| BLEU-1 | BLEU-2 | BLEU-3 | BLEU-4 | METEOR | ROUGE-L |
|--------|--------|--------|--------|--------|---------|
| 0.370 | 0.293 | 0.246 | 0.198 | 0.182 | 0.330 |

Figure 6: A sample result of the generated report and its corresponding ground truth. The generated result is relatively worse.

formance in Figure 6 is the absence of critical medical entities, such as "Periaortic Lymph Nodes". While the nearest-neighbor report helps reduce missing content, it does not fully cover all key items present in the ground truth report. How to more accurately and adaptively guide the generation model with key medical items is thus an important future work.

## 5. Conclusion

We have introduced **PathGenIC**, a multimodal in-context learning framework specifically designed for histopathology report generation. This framework generates histopathology reports by a vision language model with multimodal in-context learning. By considering contextual cues in VLMs, this approach significantly enhances VLMs' capacity to generate histopathology reports. We verify the effectiveness of this work by achieving state-of-the-art performance on the HistGen benchmark. In the future, a larger-scale evaluation can be done to demonstrate the generality of the proposed method. More elaborate in-context learning approaches will also be investigated.

## Acknowledgments

This work was funded in part by NVIDIA and the National Science and Technology Council, Taiwan, under grants 114-2425-H-006-004, 113-2622-E-006-029, 113-2634-F-006-002, and 112-2221-E-006-136-MY3.

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

## Appendix A. Detailed Prompts

In the feedback scenario, we ask GPT4o as an expert reviewer, provide it with the ground truth and the generated report, and ask it to compare two reports and provide feedback. The detailed prompt is:

You are an expert reviewer specializing in medical report quality assurance.

Ground Truth: {ground truth}

Generated Report: {generated text}

Comparing the ground truth and generated report, what is the thing that generated report lacks? what suggestion would you give to improve the content of the generated report? The suggestion should be deeply insightful. Be honest and harsh.

In the guideline scenario, we ask GPT4o as an AI analyst, provide it the several reports, and ask it to summarize the guidelines to write a report. The detailed prompt is:

You are an advanced AI analyst specializing in deep linguistic and structural analysis of medical reports.

Report 1: {report}

Report 2: {report}

...

Report 20: {report}

Deeply analyze these reports and extract habits, preferences, and especially biases that exist in reports of this category different from general TCGA reports. Your observations must be brutally insightful. Using the insights from observing the habits, preferences, and especially biases in these reports compared with other general TCGA reports. Conclude the habits, preferences, and especially biases with 5 short guidelines that are so insightful even harsh, ensuring anyone reading them knows the exact way to mimic these reports.

