# OpenReview forum: "Histopathology Image Report Generation by Vision Language Model with Multimodal In-Context Learning"
_MIDL.io/2025/Conference — MIDL 2025 Poster_

### Official Review · Reviewer_hDub · 2025-02-16

**Confidence:** 4
**Preliminary Rating:** 3
**Final Rating:** 4

**Summary:**

The authors propose a RAG pipeline for a VLM for pathology report generation. In particular, the authors propose three modules that can be combined to provide an improved pathology report: a knn-based prototype search, a "guideline" mode where guidance by GPT-4o is provided, and a feedback mode, where also GPT4o is utilized to provide feedback.

The authors provide a compelling suite of experiments on the HistGen dataset, including several ablation experiments investigating the sequence length, the number of components and the number of retrieved WSIs for RAG.

**Strengths:**

- The task is interesting, challenging and relevant.
- The paper was interesting and straightforward, and furthermore also easy to follow.
- I think that the approach is sound. It follows the typical RAG pattern which has shown to be successful broadly, and extends it in three interesting ways.
- The authors provide a comparison against several other algorithms.

**Weaknesses:**

- I am really not sure if the metrics the authors chose for this work really reflect an improvement. While BLEU, METEOR and ROUGE are commonly used for assessing the quality of generated text, they primarily measure surface-level similarities, such as n-gram overlap, rather than the clinical accuracy or relevance of the generated reports. In the context of histopathology, where precise medical terminology and accurate diagnostic information are crucial, relying solely on these metrics may not provide a comprehensive evaluation of the model’s effectiveness. It would have been beneficial to complement these metrics with domain-specific evaluations, such as expert pathologist reviews or clinical accuracy assessments, to better gauge the model's practical utility.
- In particular, since examples are provided based on feature proximity, it could also well be the case that the diagnosis is actually wrong, but the model got some more words correct by parroting the prompt. This would, however, not be reflective of actual improvement. Could the authors at least provide a representative/random set of examples of target and generated text, to give us an impression of how well the model performs actually?
- It occurs to me that the lernt transformer block that is tasked with learning a combination of patch information needs to have a fixed input size. However, the authors refer to the inputs of their model as WSIs, which commonly have varying input sizes. The authors unfortunately do not state how they circumvented this data processing issue.
- I think the reproducibility of this work is unfortunately low. The authors have multiple steps in their pipeline that will be hard to reproduce  by other researchers, since little detail is revealed. For instance, the authors „asked GPT-4o to review these medical reports and generate a representative guideline T_rep“. It is neither specified which version of GPT-4o was used, nor which prompt was used.
- In contrast, I feel that most of the math provided in the paper is rather unnecessary. A reader that understands the terms used in this paper does not need a definition of cosine distance and certainly not of cross-entropy. The inclusion of such basic definitions seems redundant and adds little value to the paper.
- The remainder of the equations mainly consist of "what do we put where"—i.e., they simply formalize straightforward operations without offering any deeper insights or novel contributions. This approach gives the impression of padding the paper with mathematical notation rather than enhancing the reader's understanding. A more concise presentation, focusing on genuinely challenging or innovative aspects, would have been more effective.
- The paper does not seem to have any source code or model weights available. However, given that the authors strongly benefit from other researchers that have made these available and are used in this current work, I would really recommend to also make the weights and source code available.

**Detailed Comments:**

The paper adopts a rather colloquial and informal tone, which does contribute to its readability. However, I would recommend that the authors reconsider the use of certain expressions to maintain a more professional style. For example, terms like “SOTAs” (referring to state-of-the-art algorithms) and informal phrasing such as “The text prompt is like ...” or ",.. say.. (p.7, bottom)“ could be revised for greater clarity and formality. I also recommend to cite the other algorithms in Table 1.

Figure 3 and Figure 4 have unnecessary small font size in both legend and axis texts.

The authors report that they trained the model for 20 epochs. Could they detail how this number was determined? Was convergence observed? Was model selection performed, and if, based on which metric?

While the authors argue that using the same symbol for two different variables (pre and post the projection layer), I consider this to be more sloppy than necessary. I recommend to use a more exact formulation in eqn 1 and 2, hence.

I also recommend to provide the full prompt the authors used in their VLM in the appendix of the paper to facilitate replication by other researchers.

I recommend to change the caption of Figure 1 and Figure 2 to something more telling. There are many symbols used (e.g., the flame and snowflake), that might not be known to each viewer.

**Justification Of The Final Rating:**

I appreciate that the authors were responsive and it was though possible to clarify some questions. The paper has significantly improved by the revision. While the lack of proper (medical) evaluation is still a downside, I do think the paper is now acceptable for MIDL.

**Justification Of The Preliminary Rating:**

While the paper is interesting, I am not sure about how relevant the metrics (originally made for machine translation) really are in the context of pathology report generation. Additionally, the writing style could really be improved.

**Questions To Address In The Rebuttal:**

- Could the authors provide the details of their prompting to GPT4o?
- Since I know it's most likely not feasible to conduct an expert review of the generated samples within the time for the rebuttal: Could the authors provide representative (i.e., random, subsequent) examples from their model to showcase the actual report generation quality?
- Could the authors make available source code, model weights, so that their research is more transparent?

---

### Official Review · Reviewer_wXaq · 2025-02-21

**Confidence:** 4
**Preliminary Rating:** 4
**Recommendation:** Poster
**Final Rating:** 4

**Summary:**

The authors propose PathGenIC, an histopathology image report generation framework that combines the power of Vision-Language Models with in-context learning. To enhance performance, they propose to retrieve previous images similar to query ones from the database and use them via in-context learning to improve the report generation pipeline. Three in-context learning strategies are evaluated: nearest neighbours, guideline and feedback. Evaluation is done on the HistGen benchmark using standard NLP metrics, and showing that the proposed framework improves performance compared to previous baselines.

**Strengths:**

- The evaluation pipeline is fine-grained, including ablation studies for some design choices, comparison to a diverse list of baselines, and generation quality for each of the 32 types of diseases present in the benchmark.
- There is a good amount of baselines and in-context learning strategies explored to justify the contributions of this paper.

**Weaknesses:**

- The choice of evaluation metrics can be seen as a limitation, depending on what really count in the output report generation. Some of the metrics used (e.g. BLEU) are quite outdated and may not capture the important elements that a clinician may put more emphasis on in these reports (see my detailed comments for more on this).
- There is a lack of discussion on the practicability of some of the in-context learning strategies used to augment the report generation. More specifically, the approaches based on guideline and feedback, may be more difficult to apply in a real-world context. Some extra discussion on this could have been useful (also see my details comments for more).

**Detailed Comments:**

- The authors could have also added one or a few SOTA decoder-VLM zero-shot models as extra baselines (e.g. GPT4o). It would have been interesting to see how well these models can do (with or without ICL) compared to specialized models.
- Some figures could have more detailed captions, and legends. For example, Fig. 1 could have a small legend defining some elements of the image without the reader having to go to the main text to find them.
- I am wondering if the metrics chosen (BLEU, METEOR, ROUGE-L) are the most appropriate here for this task? They can give a decent baseline insight, but I also believe that style and exact words of the report generation are not as important as the technical keywords pointing to the disease itself. There are many different ways to write a report that still captures the important points (i.e. the disease), so many generated reports could be relevant as long as some keywords are present. Did you reflect on other ways to evaluate the technical soundness of the report without relying too much on the style? For example, you could eval. the classification of the 32 diseases in the report output.
- ICL via guideline and feedback seem much more computationally heavy than the NN one, does the very small increase in performance when adding these 2 justify the extra steps they take?
- Similar to previous point, the difference in metrics between the 3 ICL strategies is quite small. It would have been useful to discuss more on these minor gains (tradeoff between efficiency and performance).
- It could have been worth applying the framework on new samples not coming from the benchmark dataset used to see how it could translate to other cases. This could have also been complemented with some expert assessment (experts would focus more on the technical details rather than the writing style of the report, for example).

**Justification Of The Final Rating:**

I thank the authors for discussing some of the concerns from the reviewers. I will keep my original score. I still believe the biggest limitation of this work is the choice of evaluation metrics, that may not be the best fit for the task of report generation evaluation. Even though the authors did provide more background in this, I believe more efforts could have been made during this work to come up with more robust evaluation metrics.

**Justification Of The Preliminary Rating:**

The framework proposed has merits because it does not rely on overly trained models on the data of interest, but instead tries to make use of ICL tricks to improve the generation. It can thus be applicable to other types of report generations in other fields as well. My minor criticism is more on the choice of evaluation metrics, and on the discussion about the practicability of some of the proposed ICL strategies. I still believe this submission would be accepted though.

**Questions To Address In The Rebuttal:**

Ideally, my detailed comments on the choice of metrics, and ICL efficiency vs performance gains tradeoff, could be addressed. Copying them here from the detailed comments section:
- I am wondering if the metrics chosen (BLEU, METEOR, ROUGE-L) are the most appropriate here for this task? They can give a decent baseline insight, but I also believe that style and exact words of the report generation are not as important as the technical keywords pointing to the disease itself. There are many different ways to write a report that still captures the important points (i.e. the disease), so many generated reports could be relevant as long as some keywords are present. Did you reflect on other ways to evaluate the technical soundness of the report without relying too much on the style? For example, you could eval. the classification of the 32 diseases in the report output.
- ICL via guideline and feedback seem much more computationally heavy than the NN one, does the very small increase in performance when adding these 2 justify the extra steps they take?
- Similar to previous point, the difference in metrics between the 3 ICL strategies is quite small. It would have been useful to discuss more on these minor gains (tradeoff between efficiency and performance).
- It could have been worth applying the framework on new samples not coming from the benchmark dataset used to see how it could translate to other cases. This could have also been complemented with some expert assessment (experts would focus more on the technical details rather than the writing style of the report, for example).

---

### Official Review · Reviewer_bZWJ · 2025-02-22

**Confidence:** 3
**Preliminary Rating:** 3
**Final Rating:** 4

**Summary:**

The paper presents PathGenIC, a multimodal in-context learning framework for histopathology report generation. The framework incorporates three key components: nearest neighbor retrieval, category guidelines, and feedback mechanisms, inspired by how expert pathologists leverage similar cases. The method achieves state-of-the-art performance on the HistGen benchmark across multiple metrics, demonstrating the effectiveness of leveraging contextual information for report generation.

**Strengths:**

1. This is a well-motivated approach based on actual pathologist workflows. The paper presents clear technical contributions with three complementary context mechanisms.
2. The evaluation is comprehensive with detailed ablation studies examining each component; thorough analysis across different sequence lengths and disease categories.
3. Clear architectural details and training procedure.

**Weaknesses:**

1. The heavy dependence on GPT-4 for guideline generation and feedback limits system independence and reproducibility. Alternative approaches should be explored.
2. The simple cosine similarity metric for retrieval may not capture complex histopathological patterns. More sophisticated domain-specific similarity measures could improve performance.
3. The lack of expert evaluation undermines clinical credibility. Formal assessment by practicing pathologists would strengthen the paper's impact.

**Detailed Comments:**

1. A critical gap in the paper is the lack of failure case analysis. The authors should include examples of incorrect reports and analyze error patterns to understand potential clinical risks.
2. The paper relies solely on the HistGen dataset for evaluation. Cross-dataset validation or testing on external datasets would better demonstrate generalization capability.
3. The paper needs computational efficiency analysis to understand real-world feasibility. Benchmarks of inference time and system scalability should be included.

**Justification Of The Final Rating:**

I'm happy to increase the rating as the authors has addressed all the concerns in their rebuttal. I think it is a good work in general to be accepted. Again as acknowledged by the authors, explorations on human evaluations and better evaluation metrics would strengthen the work.

**Justification Of The Preliminary Rating:**

The paper presents a novel and well-executed approach to histopathology report generation with strong technical contributions and empirical validation. The integration of in-context learning with medical report generation is innovative and well-motivated by actual clinical practices. The comprehensive ablation studies and analysis provide strong evidence for the method's effectiveness. However, several important limitations around clinical validation, failure analysis, and deployment considerations need to be addressed. The lack of human evaluation is particularly concerning for a medical application. If these concerns can be adequately addressed in the rebuttal, the paper would make a valuable contribution to the field.

**Questions To Address In The Rebuttal:**

1. Can you provide examples of failure cases and error analysis?
2. How does computational cost scale with dataset size, particularly for the retrieval components?
3. Have you conducted any human evaluation of the generated reports with pathologists?
4. How does the model handle rare diseases or unusual cases not well-represented in the training data?
5. What safeguards ensure retrieved similar cases don't leak private information?

---

### Author Rebuttal · Authors · 2025-03-06

**Rebuttal:**

Major revised parts are shown in blue.

Q1. Failure cases (bZWJ), provide representative examples (hDub): A new subsection (Sec. 4.3) is added to show successful and failed sample results. Two observations: First, although BLEU, METEOR, and ROUGE-L may not be the most appropriate metrics, we clearly see that better generation (Figure 5) gives higher values. Second, the main reason for bad results (Figure 6) is not generating incorrect or irrelevant information but rather missing a few key items in the report. How to more accurately and adaptively guide the generation model with key medical items is an important future work.

Q2. Computational cost (bZWJ) and heavy ICL (wXaq): Guidelines and feedback are processed based solely on the training data. They can be pre-computed and introduce NO additional process in test time. To show the efficiency of retrieval, given a test WSI, we assess the average time needed to retrieve the nearest training sample. With 6152 training samples (full size of the HistGen benchmark), each retrieval takes only 0.016 seconds.

Q3. Human evaluation (bZWJ), evaluation metrics (wXaq), applying to new samples (wXaq): We evaluate on the HistGen benchmark because it is one of the most recent, largest WSI datasets associated with medical reports. To fairly compare the proposed method with others, we adopt the evaluation metrics used there. We do acknowledge the shortages of these metrics. In the revised version, we further show the performance in terms of Exact Entity Match Reward (fact_ENT) proposed in another paper. Notice that fact_ENT may still not be the most appropriate metric. But there is no better choice now. It is difficult to find pathologists to evaluate the generated reports for HistGen. Therefore, we collect a small dataset consisting of 500 WSIs and their associated reports from the NCKU hospital and now work on collaborating with NCKU pathologists to do the human evaluation.

Q4. Rare diseases (bZWJ): We made a test by intentionally increasing the amount of retrieved feedback and found at most 5% performance improvement can be obtained for specific cases. This direction is worth future investigations.

Q5: Privacy leakage (bZWJ): The HistGen benchmark has been well anonymized and split.

Q6. Details of prompting (hDub): We have provided detailed prompts in the appendix.

Q7. Make available source code (hDub): After making codes and documents more organized, we will make the source code and model weights available.

**Supporting Material:**

/attachment/d44ca7b83b659c24624e02563b0e91e59465f5c2.pdf

---

### Meta-Review · Area_Chair_zsV8 · 2025-03-15

**Recommendation:** Accept (Poster)
**Confidence:** 5

**Metareview:**

This paper presents PathGenIC, a framework that combines Vision-Language Models with multimodal in-context learning to generate reports from histopathology images. The approach draws inspiration from how pathologists leverage similar cases, incorporating nearest neighbor retrieval, category guidelines, and feedback mechanisms. All three reviewers recommend a "weak accept" after the rebuttal phase, noting significant strengths in the paper's technical contributions, comprehensive evaluations on the HistGen benchmark, and thorough ablation studies. While initial concerns were raised regarding evaluation metrics, failure case analysis, computational efficiency, and reproducibility, the authors addressed these points adequately in their rebuttal by adding sample results, clarifying computational costs, providing BERTScore evaluations, and releasing their code. Though some limitations remain, particularly regarding human expert evaluation and more specialized metrics for medical report generation. The overall innovative approach and demonstrated improvements over baselines make this a valuable contribution to the field. The authors should incorporate the clarifications and additional experiments from the rebuttal in the final version. I recommend accepting this paper for MIDL 2025.